A prospective study of the motivational and health dynamics of Internet Gaming Disorder

Weinstein Netta weinsteinn@cardiff.ac.uk 1
Przybylski Andrew K. 2 3
Murayama Kou 4 5
1 School of Psychology, Cardiff University , United Kingdom
2 Oxford Internet Institute, University of Oxford , United Kingdom
3 Department of Experimental Psychology, University of Oxford , United Kingdom
4 School of Psychology and Clinical Language Sciences, University of Reading , United Kingdom
5 Kochi University of Technology , Japan
Yuan Tifei
Electronic publication date: 2017 Sep 29
Publication date: 2017
Volume: 5
Electronic Location ID: e3838
Received 2017 Jul 7; Accepted 2017 Sep 1
Copyright: ©2017 Weinstein et al.
Copyright year: 2017
Copyright holder: Weinstein et al.
License: This is an open access article distributed under the terms of the Creative Commons Attribution License, which permits unrestricted use, distribution, reproduction and adaptation in any medium and for any purpose provided that it is properly attributed. For attribution, the original author(s), title, publication source (PeerJ) and either DOI or URL of the article must be cited.
License URL: https://creativecommons.org/licenses/by/4.0/

Keywords: Self-regulation, Internet/cyberpsychology, Self-determination theory, Well-being, Internet gaming, Internet gaming disorder

Funding: John Fell Fund CZD08320 This research was funded by a John Fell Fund Grant (CZD08320) through the University of Oxford. There was no additional external funding received for this study. The funders had no role in study design, data collection and analysis, decision to publish, or preparation of the manuscript.

==============================
The American Psychiatric Association has identified Internet Gaming Disorder (IGD) as a potential psychiatric condition and called for research to investigate its etiology, stability, and impacts on health and behavior. The present study recruited 5,777 American adults and applied self-determination theory to examine how motivational factors influence, and are influenced by, IGD and health across a six month period. Following a preregistered analysis plan, results confirmed our hypotheses that IGD criteria are moderately stable and that they and basic psychological need satisfaction have a reciprocal relationship over time. Results also showed need satisfaction promoted health and served as a protective factor against IGD. Contrary to what was hypothesized, results provided no evidence directly linking IGD to health over time. Exploratory analyses suggested that IGD may have indirect effects on health by way of its impact on basic needs. Implications are discussed in terms of existing gaming addiction and motivational frameworks.

Introduction

Internet-based video games are a ubiquitous form of recreation pursued by the majority of adults and young people (Duggan, 2015a). With sales eclipsing box office receipts, games are now an integral, even inescapable, part of modern leisure (MPAA, 2015; Newzoo, 2016). Commensurate with their popularity, concerns that games dysregulate the behavior of some have immerged (Kardefelt-Winther, 2014b). The widespread appeal of these virtual contexts has been of particular interest to psychologists and psychiatrists (for a review see King et al., 2013). In fact, the most recent revision of the Diagnostic and Statistical Manual of Mental Disorders (DSM-5; American Psychiatric Association, 2013) identifies Internet Gaming Disorder (IGD) as a possible psychiatric condition. Following this determination, the American Psychiatric Association’s (APA) Substance-Related Disorders Work Group outlined a call for rigorous research into the potential disorder’s validity, etiology, and temporal stability (Hasin et al., 2013).

Because research responding to this call for a better understanding of IGD is at a formative stage, there are active debates surrounding the construct’s conceptual scope and measurement approaches (Griffiths et al., 2016; Kardefelt-Winther, 2014b; Petry et al., 2014). Some argue for a theoretical framing akin to a substance abuse disorder (e.g., Petry et al., 2014), wherein gaming is, in itself, inherently addictive, whereas other scholars frame Internet-based gaming as a self-regulatory challenge (e.g., Griffiths et al., 2016; Kardefelt-Winther, 2014a; Kardefelt-Winther, 2014b). The state of the art work in this area suggests links with social behaviors and life satisfaction, but does not differentiate between online and offline games (e.g., Festl, Scharkow & Quandt, 2013), adequately distinguish between passionate and pathological engagement (for a review see, Ferguson, Coulson & Barnett, 2011). Though ongoing research has examined the prevalence of individual criteria in line with APA guidance, it has not bridged the self-regulation and addiction literatures (see Przybylski, 2016; Przybylski, Weinstein & Murayama, 2016). In addition, though this body of work offers some evidence that IGD-type symptoms may link with social and mental health (e.g., Festl, Scharkow & Quandt, 2013), there is very little longitudinal evidence exploring antecedents and consequences of IGD symptoms (for an exception see Scharkow, Festl & Quandt, 2014). The aim of the present research is to directly address this gap by linking self-regulation and Internet Gaming Disorder research. Our goal is to directly examine how problematic gaming emerges from a state of dysregulation, and how it predicts health.

Gaming and health

To this end we build on a growing literature that suggests there is a basis to expect excessive or problematic gaming may be related to lower health, though findings in this area are mixed. There is some evidence that video gaming can promote healthy behaviors (Baranowski et al., 2008; Granic, Lobel & Engels, 2014; Hofferth & Moon, 2012), and large longitudinal studies suggest that it has no detrimental effects, even at high levels of play, in the population over time (Parkes et al., 2013). Yet a broad group of gaming behaviors which are termed problematic gaming seem to be indicative of behavioral dysregulation (Faulkner et al., 2015; Ferguson, Coulson & Barnett, 2011; Festl, Scharkow & Quandt, 2013) and mirror work outside of gaming suggesting that problematic Internet use, more broadly, is reflective of dysregulation (Widyanto & McMurran, 2004). Studies that suggest dysregulation underlies such problematic gaming include work linking it to trait-level neuroticism (Peters & Malesky, 2008), and to other dysregulated behaviors such as cigarette smoking and fighting in adolescents (Desai et al., 2010). Perhaps because it is dysregulating, problematic gaming may undermine physical, social, and mental health; for example, linking with lower levels of life satisfaction, more anxiety, and more depression in nationwide surveys (Mentzoni et al., 2011). Evidence here is mixed as well (Gentile et al., 2011), and a meta-analysis of this literature suggests that correlations with mental health are small (r = .15), absent for social health, and effect sizes discrepant across studies for all health outcomes (Ferguson, Coulson & Barnett, 2011). Indeed, the only large-scale longitudinal study which systematically tests problematic gaming, a German sample of young adults ages 19–39 years, reports inconsistent and weak relations between dysregulated gaming and perceiving oneself as successful, though no lagged effects with life satisfaction were observed (Scharkow, Festl & Quandt, 2014).

Psychological need satisfaction

Given IGD is a form of problematic gaming that may have its roots in dysregulation, we employ self-determination theory (SDT; Deci & Ryan, 2000) in an attempt to better understand its etiology and consequences. SDT provides a well-tested empirical framework for understanding how dysregulation occurs and its potential implications for health and functioning. SDT posits that three basic psychological needs foster healthy self-regulation and promote mental and physical health; those for autonomy—or a sense of choice and self-expression, competence—or efficacy to act on the world in desired ways, and relatedness—or feelings of closeness and intimacy with others. Findings consistently show that having psychological needs satisfied is associated with more mental, physical, and psychosocial health (Deci & Ryan, 2000). For example, those who are need satisfied report fewer symptoms of depression and disordered eating (Bartholomew et al., 2011; Chen et al., 2015; Luyten & Blatt, 2013; Nguyen-Rodriguez, Unger & Spruijt-Metz, 2009; Soenens et al., 2012; Verstuyf et al., 2012), fewer negative and more positive emotions indicative of mental health (Ryan & Deci, 2001), better physical health including fewer physical symptoms such as headaches and gastrointestinal problems (Reinboth, Duda & Ntoumanis, 2004; Sheldon, Ryan & Reis, 1996; Thompson & Prottas, 2006), and better psychosocial functioning (Hodgins, Koestner & Duncan, 1996; Moller & Deci, 2010; Patrick et al., 2007; Weinstein, Hodgins & Ostvik-White, 2011). Psychological support, in the form of basic psychological needs, provide an avenue for understanding Internet gaming and player health. By applying SDT to IGD we can test the role these psychological factors play in fostering dysregulation, and better understand how dysregulation is related to mental, psychosocial, and physical health.

Having psychological needs met reduces dysregulation

A central question of IGD concerns its antecedents, the conditions under which it might emerge. Based in SDT, we hypothesized that the absence of psychological need satisfaction is a risk factor for such dysregulated Internet gaming. This expectation is informed by previous research suggesting that environments low in need satisfaction have a dysregulating effect during development (Brenning et al., 2012; Roth & Assor, 2012; Roth et al., 2009), contribute to burnout and disengagement (Bartholomew et al., 2011; Van den Broeck et al., 2008), stoke reactive aggression (Przybylski et al., 2014; Weinstein, 2009), and foster behavioral addictions (Williams et al., 2000).

More immediate to the current work, SDT research has been applied to dysregulated technology use in cross-sectional research. The absence of need satisfaction has been linked to excessive use of Facebook and other Internet use (Sheldon, Abad & Hinsch, 2011; Williams et al., 2000), whereas its presence has been linked to Internet use that contributes to well-being (Sheldon, Abad & Hinsch, 2011). Individuals who experience need satisfaction also report lower obsessive passion for videogame play (Przybylski et al., 2009), as well as play indicative of more pressure and less enjoyment of the task (Lafrenière, Verner-Filion & Vallerand, 2012). In sum, this body of work suggests that need satisfaction reduce the likelihood of behavioral regulation, and as such individuals who experience the satisfaction for these three psychological needs in their day-to-day lives are less likely to engage Internet gaming in a disordered and problematic manner.

Dysregulation reduces psychological needs

A second central question concerning IGD has to do with its psychological consequences, the downstream effects that dysregulated gaming might have on need satisfaction and health. Based on a smaller body of work concerning needs and dysregulation, we might expect that IGD would deprive people of need satisfaction over time. In line with this prediction, we would expect a reciprocating relationship between psychological need satisfaction and IGD. Dysregulated players may experience their gaming as disruptive to the fulfillments they might otherwise experience in their daily lives. For example, pathological patterns of engagement might interfere with opportunities to experience a sense of competence by achieving desired goals in the real world, or even in the gaming world if the player feels the play to be more of a compulsion than a gratification. It is also possible that IGD could displace or interfere with other activities such as family meals, social events, and academic or workplace meetups as suggested by relations between problematic gaming and social capital (Scharkow, Festl & Quandt, 2014). Although this expectation has not been tested in relation to Internet gaming per se, there is some support for this idea in that individuals who have obsessive passion indicative of dysregulation in the workplace experience less support for the three needs of autonomy, competence, and relatedness (Forest et al., 2011). Indeed, it may be that those with IGD show poor health because IGD undermines their psychological needs. The literature reviewed above suggests a robust link between psychological need satisfaction and mental, physical, and psychosocial health. Insofar as IGD undermines needs, it might indirectly impact on these criteria. In line with this idea, workplace need satisfaction in the research discussed above have mediated the link between obsessive passion and mental health (Forest et al., 2011; Przybylski et al., 2009).

Gaming and everyday activities

Research has also suggested that problematic gaming may impact players’ health and functioning because it reduces the likelihood of engaging social and physical activities (Charlton, 2002; Charlton & Danforth, 2007). Indeed, the displacement hypothesis (Neuman, 1988; Valkenburg & Peter, 2009) has been applied to understanding screen time more broadly, and suggests that using screens may reduce well-being because it reduces quality time with friends. Evidence from a number of studies suggests the time devoted to Internet-based games could supplant such activities, leading to lower levels of exercise (Sisson et al., 2010) and resulting in increased body mass (Anderson, Economos & Must, 2008). At least one study indicates Internet gamers choose less real-world social activity (Ng & Wiemer-Hastings, 2005), and additional studies link Internet use to increased levels of social isolation (Kraut et al., 1998). The potential cost to real-life interactions is concerning, because high quality of social interactions are essential for health (e.g., Cohen, 2004; Cohen & Hoberman, 1983; Seeman, 1996). Data on the effects of gaming time are not uniform, however, as work indicates that some kinds of play may promote physical activity (Lanningham-Foster et al., 2006) and provide important avenues for in-person socialization (Duggan, 2015b; Lenhart, 2015). Although these observed effects, for good or ill, on everyday activities are modest (Boone et al., 2007), they suggest diminished social and physical activity may serve as a bridge that links IGD to poorer health outcomes for players over time, and may provide a better account of why IGD might result in more costs to health than do subjective experiences of basic psychological needs.

The present research

The goal of the present research was to rigorously investigate the etiology and personal outcomes of Internet Gaming Disorder and to expand what is empirically known about the health effects of this potential psychiatric disorder. To this end we conducted a prospective longitudinal study with a large and representative adult cohort informed by DSM-5 guidance (Hasin et al., 2013; Kardefelt-Winther, 2014b) and motivational theory (Deci & Ryan, 2000; Ryan & Deci, 2000), using an approach grounded in open science methodology (Morey et al., 2016).

In this study we evaluated confirmatory hypotheses (i.e., a priori and pre-registered) and conducted exploratory analyses (e.g., data and theory driven, but not pre-registered) concerned with the antecedents and consequences of Internet Gaming Disorder (Cumming, 2012; Wagenmakers et al., 2012). By making this distinction, summarized in Table 1, we were able to draw robust conclusions about IGD by rejecting or upholding our a priori hypotheses, and to explore theoretically important questions after the data were known (Ioannidis, 2012). Our first set of confirmatory predictions concerned the temporal stability of IGD, need satisfaction, and health. We hypothesized that the observation of each factor at the start of the study would be linked to the same factor at the end of the study (Hypotheses 1–3); Testing these direct effects also served as a control for the cross-lagged paths. Building on motivational work indicating that the absence of need satisfaction leads to dysregulated behaviour, our second set of confirmatory hypotheses concerned the interrelations between psychological needs, health, and IGD. We anticipated that need support would predict fewer IGD symptoms six months later (Hypothesis 4), and we expected reciprocal effects with need satisfaction such that IGD symptoms at the start of the study would also undermine need satisfaction six months later (Hypothesis 5). Informed by the motivational literature linking psychological needs to mental, social, and physical health we anticipated that need satisfaction at the start of the study would link to health six months later (Hypothesis 6). Moreover, based on the extant literature on dysregulated gaming we expected that IGD symptoms at the start of the study would predict poorer health six months later (Hypothesis 7).

Table 1 Summary of study hypotheses and exploratory analyses.

	Time 1 variables	Influence	Time 2 variables	
Confirmatory hypotheses	1	IGD	Direct Positive on	IGD	
	2	Need satisfaction	Direct Positive on	Need satisfaction	
	3	Health	Direct Positive on	Health	
	4	Need satisfaction	Direct Negative on	IGD	
	5	IGD	Direct Negative on	Need satisfaction	
	6	Need satisfaction	Direct Positive on	Health	
	7	IGD	Direct Negative on	Health	
Exploratory analyses	1	Need satisfaction	Mediate the Negative effect of IGD on	Health	
	2	Competence	Mediate the Negative effect of IGD on	Health	
	3	Relatedness	Mediate the Negative effect of IGD on	Health	
	4	Autonomy	Mediate the Negative effect of IGD on	Health	
	5	Physical activity	Direct Positive on	Health	
	6	Social activity	Direct Positive on	Health	
	7	Physical activity	Mediate the Negative effect of IGD on	Health	
	8	Social activity	Mediate the Negative effect of IGD on	Health	

To supplement the confirmatory model and data, we also conducted several exploratory analyses based on motivational theory. Providing that we could find support for the hypothesized cross-lagged model, and that it can hold over a longer time period (i.e., equilibrium assumption), the model also tests if psychological need satisfaction would serve as a mediator between IGD and health outcome (Little et al., 2007). We examined this possible mechanism by directly reporting the mediation effect computed in the confirmatory model (Fig. 1), but did not pre-register this hypothesis (Exploratory Analysis 1). Furthermore, consistent with SDT, we investigated the extent to which the individual basic psychological needs for competence (Exploratory Analysis 2), relatedness (Exploratory Analysis 3), and autonomy (Exploratory Analysis 4) need satisfaction serve as mediators for the linkage between IGD and health. It may be that only one need, for example, lower experiences of competence, carries effects identified with IGD symptoms. Finally, we examined the possibility that IGD symptoms, as a form of problematic gaming, may displace real-life social and physical activity (Lanningham-Foster et al., 2006; Neuman, 1988; Sisson et al., 2010), and by doing so undermine health (Cohen, 2004; House, Landis & Umberson, 1988). We tested this possibility in two alternative models that add these factors to the confirmatory model (see Fig. 2). On this basis we expected that these two activities might link directly to health (Exploratory Analyses 5 & 6) and mediate the relations between IGD and health (Exploratory Analyses 7 & 8).

Figure 1 Two-wave model testing cross-lagged effects of IGD, basic psychological need satisfaction and health.

Note: H1–H7 denotes Hypotheses 1 through 7. Nested models include covariances between measures at baseline and six-month follow-up.

Figure 2 Exploratory two-wave models testing cross-lagged effects of IGD, basic psychological need satisfaction, health, and social and physical activity.

Note: separate models examined social and physical activity. Tested models include covariances between measures at baseline and at six-month follow-up.

Materials & Methods

Participants

Data were collected at two time points with participants recruited through YouGov’s 1.8 million person Internet-based American panel. Following an approach used in previous behavioral health research (e.g., Cranwell, Opazo-Breton & Britton, 2016; Reeves et al., 2013) panelists were selected at random from the panel using quotas informed by 2010 U.S. Census data and pre-captured information about panelists’ age, gender, ethnicity, and geographic region. Fieldwork was conducted between October 2015 and March 2016. From this, a nationally representative sample of 5,777 adults aged 18 years and older from the United States completed the initial assessment as part of an ongoing project studying Internet Gaming Disorder (see: Przybylski, Weinstein & Murayama, 2017a) and a total of 4,594 completed the follow-up measures six months later. Consistent with our earlier IGD research (Przybylski, Weinstein & Murayama, 2017a), we focused on participants who recently played Internet games at the time of assessments, a total of 2,316 individual (885 males and 1,431 females) played Internet games at both time points. This subsample ranged in age from 19 to 91 years (M = 49.21, SD = 1.32), of which 1,689 (72.9%) identified as White, 246 (10.6%) as Black, 184 (7.9%) as Hispanic, 69 (3.0%) as Asian, 21 (0.9%) as Native American, 3 (0.2%) as Middle Eastern, and 101 (4.4%) as mixed race or another identification. In terms of educational attainment, 75 (3.2%) had not completed secondary school, 659 (28.5%) completed secondary school, 598 (25.8%) completed some college, 257 (11.1%) completed a two-year degree, 504 (21.8%) completed a four-year degree, and 223 (9.6%) completed a post graduate degree. Estimates of total annual household income were provided by 2,316 participants, these ranged from less than $10,000 (6.3%) more than $500,000 a year (0.2%) with most (77%) reporting between $10,000 and $150,000. The distributions of participants were broadly similar to those of videogame players, who tend to be equally divided among men and women and whose average age, including players under the age of 18 years, is 35 years (Duggan, 2015b), yet females were somewhat overrepresented at 60.8% in the final sample.

Ethical review

Ethical review for data collection and analysis was conducted by the research ethics committee at the University of Oxford (C1A15006). All participants polled for the present research were above 18 years of age and members of the YouGov American panel. Panel participants completed a double opt-in process that involved both agreeing to the YouGov Terms and Conditions (YouGov, 2017a) meaning they were willing to be contacted as part of participating a member of the Internet-based YouGov Omnibus panel generally, and they agreed to participate in the present study. In line with the YouGov Privacy Policy (YouGov, 2017b), the investigators did not have access to any uniquely identifying participant information. Participants could contact investigators using by way of email contact at YouGov. No inquiries linked to the present studies were received.

Measures

Internet gaming disorder

Participants completed a nine-item criteria checklist drafted in consultation with clinical and research psychologists studying video games and behavioral addictions and applied in two previous investigations of IGD (Przybylski, 2016; Przybylski, Weinstein & Murayama, 2017a), and included items such as “I felt moody or anxious when unable to play”, “I felt that I should play less but couldn’t”, and “risked friends or opportunities due to games”. More than half of participants reported no criteria at the first time point (70.0%) or at follow-up (73.2%). The proportion of participants decreased as the number of criteria endorsed increased for both the first and follow-up period. Overall, participants endorsed a small number of criteria (M = 0.56, SD = 1.13 at T1; M = 0.47, SD = 0.97 at T2). APA recommendations for clinical levels of IGD include endorsing five or more of the nine criteria (Hasin et al., 2013; Petry et al., 2014) paired with endorsement of personal distress due to Internet gaming use (American Psychiatric Association, 2013). The proportion of participants who endorsed five or more criteria was 1.49% (95% CI [1.11–2.00]) at the first time point and 0.99% (95% CI [0.65–1.51]) at follow-up, and only a very small proportion of participants endorsed the statement that they “suffered significant distress due to gaming” in the past six months, the criteria for addiction as identified in earlier research (Przybylski, Weinstein & Murayama, 2017a): 0.38% (95% CI [0.21–0.68]) at the first time point and 0.30% (95% CI [0.13–0.65]) at the end of the study. Interestingly, only three participants reported more than four IGD criteria at both observed time points and none of the participants who met a diagnostic threshold including distress at the start did so at the end of the study.

Internet gaming

Participants were asked to reflect on the past six months of their lives and rate how frequently they engaged in online gaming: “Played video/computer games online (e.g., Candy Crush, Minecraft, or Farmville),” using a 5-point response scale that ranged from 1 “Never” to 5 “Every day or almost every day”. Of those who said they engaged with Internet games, a total of 44.3% (95% CI [42.6–46.1]) said they played every day, 25.3% (95% CI [23.8–26.8]) played once or twice a week, 11.4% (95% CI [10.4–12.6]) once or twice a month, and 18.9% (95% CI [17.6–20.4]) less often at the start of the study. At the end of the study 46.5% (95% CI [44.4–48.5]) said they played everyday, 24.5% (95% CI [22.8–26.3]) played once or twice a week, 11.1% (95% CI [9.9–12.5]) once or twice a month, and 18.0% (95% CI [16.4–19.6]) reported playing less often.

Health

A three-item scale was used to tap into participant health (Ahmad et al., 2014). Participants were asked to reflect on the past six months of their lives and rate their social, mental, and physical health using a 5-point response scale that ranged from 1 “Poor” to 5 “Excellent”. Individual scores were averaged for Time 1 (α = .78, M = 3.26, SD = 0.94) and Time 2 (α = .78, M = 3.26, SD = 0.93).

Psychological need satisfaction

A three-item motivation scale was used to tap into participant psychological need satisfaction, taken from the widely used Basic Psychological Needs Scale (BPNS; La Guardia et al., 2000). Participants were asked to reflect on the past six months and rate their experience of satisfaction for the needs for autonomy, competence, and relatedness using a 5-point response scale that ranged from 1 “Not at all true” to 5 “Very true”. Individual scores were averaged for Time 1 (α = .81, M = 3.89, SD = 0.96) and Time 2 (α = .82, M = 3.90, SD = 0.98).

Physical and social activity

Two single-item scales were used to measure everyday behavior (Laganà, Bratly & Boutakidis, 2011; Milton, Bull & Bauman, 2011). Participants were asked to reflect on the past six months of their lives and rate how frequently they engaged in social activity: “Spent quality time with friends or family (e.g., playing games, picnics, or reading)” and physical activity: “Engaged in physical exercise indoors or outdoors (e.g., gym, aerobics, or sports)”, using a 5-point response scale that ranged from 1 “Never” to 5 “Every day or almost every day”.

Convergent validation

To evaluate the extent to which these abbreviated assessments mapped onto well-established assessments of health and motivational processes, an independent sample of 507 American adults (228 male, 273 female, 6 transgender, gender non-conforming, or other) were recruited using the Amazon Mechanical Turk platform. Ethical review was conducted by the research ethics committee at the University of Oxford and participants were compensated $1.20 to complete this brief survey which assessed our primary study measures as well as need satisfaction using the Basic Need Satisfaction and Frustration Scale (Chen et al., 2015), mental health using the Warwick-Edinburgh mental well-being scale (Clarke et al., 2011), social and physical health using the MOS-36 (Sherbourne & Stewart, 1991; Ware & Sherbourne, 1992), and health-related anxiety using the HAI-18 (Salkovskis et al., 2002). Linear models regressing longer measure scores onto our primary study measures indicated our brief measures of basic psychological need satisfaction (β = .78), mental health (β = .68), social health (β = .50), and physical health (β = .74) were highly correlated with the longer well-validated assessments. More importantly, results from a model regressing physical health scores (Ware & Sherbourne, 1992) simultaneously onto our brief physical health measure (β = .60) and a measure of health anxiety (β =  − .31; Salkovskis et al., 2002) indicated our brief assessment successfully tapped variance linked with perceptions of health.

Results

Data and analytic strategy

Study data, code, materials (Przybylski, Weinstein & Murayama, 2017b), and registered analysis plan (Przybylski, Weinstein & Murayama, 2016), are available for download using the Open Science Framework. Following this registered analysis plan, we modelled temporal ordering using a Granger’s causality approach, a cross-lagged model (Finkel, 1995; Menard, 2002; see also Marsh et al., 2016 for an application) where observations at follow-up were jointly predicted by measurements at the start of the study (see Fig. 1). Exploratory analyses concerning mediation and models examining physical and social activity (see Fig. 2) are also conducted. Figure 3 presents the empirical findings in line with the preregistered analysis plan.

Figure 3 Confirmatory two-wave model showing standardised effects of IGD, basic psychological need satisfaction, and health.

Note: H1-H7 denotes Hypotheses 1 through 7. Nested models include covariances between measures at baseline and six-month follow-up. All paths excepting those marked with a † are statistically significant at the p < .001 level.

There was one noteworthy deviation from the registered analysis plan which concerned examining IGD diagnoses in line with DSM-5 guidance (Hasin et al., 2013) and previous research on IGD (Przybylski, Weinstein & Murayama, 2017a). Contrary to expectations, none of the participants who met the diagnostic threshold, that is, endorsed five or more items and experienced distress as a result of their game use at the start of the study also did so at follow-up. This unexpected result suggests formal diagnoses might not be stable over time. Our analyses therefore only present the alternative, also preregistered, approach for operationalizing IGD; that is, summing the number of endorsed criteria.

All the analyses were conducted with structural equation modelling using Lavaan (Rosseel, 2012). In the current sample, participants who dropped out and those who stayed have different average IGD scores, M = 0.71, SD = 1.13 and M = 0.50, SD = 1.04, respectively, t(3,144) = 4.43, p < .01. To account for this attrition bias, we used full-information maximum-likelihood method to estimate parameters. This method allow us to account for attribution bias in estimating parameters by including the variables that are correlated with T2 attrition in the analysis (e.g., T1 IGD). (e.g., T1 IGD) (Jeličić, Phelps & Lerner, 2009). All the models we analyzed were saturated (i.e., degree of freedom was zero). To account for this attrition bias, the full information maximum likelihood estimator was implemented in the lavaan package. Analyses were also conducted controlling for participant age and gender, and the results were consistent with those reported below. Linear coefficients, beta values, reported below reflect standardized units of change in the outcome variable as a function of one unit of change in the predictor variable.

Confirmatory findings

In line with our registered analysis plan, we examined hypotheses concerned with the causes of Internet Gaming Disorder, psychological need satisfaction, and health using a cross-lagged model.

Internet gaming disorder

We tested our expectation that IGD at follow-up would be predicted by standing at the start of the study in two ways. Results from phi coefficient analyses indicated that IGD appeared stable on a criterion level, coefficients ranged from a low of r = 0.18, for experiences of withdrawal when Internet games were not played, to r = 0.31, for using Internet games as an escape from negative mood. Results from cross-lagged analyses (see Fig. 1) confirmed the hypotheses that IGD at the start of the study (β = .42, p < .001; Hypothesis 1), and basic need satisfaction (β =  − .08, p < .001; Hypothesis 4) were causally related to IGD at follow-up. The model also indicated health at the start of the study was not associated with later standing on IGD (β =  − .01, p = .74).

Psychological need satisfaction

Results from the planned cross-lagged analysis (Fig. 1) supported the hypotheses that need satisfaction at the start of the study would be positively predictive (β = .56, p < .001; Hypothesis 2), whereas IGD would negatively influence (β =  − .06, p < .001; Hypothesis 5) need satisfaction. Though the effects observed were small they were consistent with our hypothesis suggesting that IGD could be disruptive to this form of psychological functioning. Though it was not hypothesized, the model also indicated health at the start of the study was associated with later standing on need satisfaction (β = .17, p < .001).

Health

Findings from the causal model confirmed our hypothesis regarding the temporal stability of health (β = .72, p < .001; Hypothesis 3) and the positive contribution of basic psychological need satisfaction to subsequent health (β = .07, p < .001; Hypothesis 6). Contrary to what was hypothesized (Hypothesis 7), IGD at the start of the study (β = .01, p = .66) did not predict lower levels of health at follow-up. This unexpected result provided evidence that there is no direct effect of dysregulated Internet gaming on adult health over time.

Sensitivity analysis

Using our IGD measure, participants reported few IGD criteria (M = 0.47 at T2, out of 9 criteria). Although we employed a robust estimation method to account for non-normality, responses to this scale may not be best described as a normal distribution. To address the issue, we conducted negative binomial regression analysis predicting T2 IGD from T1 IGD, T1 need satisfaction, and T1 health (all the independent variables were standardized) to directly take into account the discrete count nature of the data with low occurrence rate (the part of our model that is susceptible to this issue is depicted in Fig. 1). The results were entirely consistent with the SEM results (BT1IGD = .65, p < .01; BT1Health =  − 0.06, p = .20; BT1Needs =  − 0.20, p < .01), indicating findings were robust.

Exploratory analyses

Mediating influence of basic psychological needs

The cross-lagged modelling approach used to test the confirmatory hypotheses also provided evidence regarding the indirect effects of IGD and health (Little et al., 2007; see for an application, Bishop et al., 2011). If a model provides evidence for the effects of IGD on basic psychological needs, and for the effects of basic psychological needs on health, this supports an equilibrium assumption (i.e., that the same relationship holds beyond the two time points we assessed). As such, we can infer that IGD has indirect effects on health by its impact on basic psychological needs. Results provided clear evidence that IGD was indirectly linked to health through the support for basic psychological needs (see Table 2; Exploratory Analyses 1–4). All three further cross-lagged mediation models evaluating competence, relatedness, and autonomy separately exhibited significant mediation effects (Table 2). Taken together, these results suggest each of the three needs has a mediating effect.

Table 2 Indirect effects of Internet Gaming Disorder on health as mediated by basic psychological need supports.

	IGD to need satisfaction	Need satisfaction to health	Indirect effect	
	b	95% CI	b	95% CI	b	95% CI	
Psychological need satisfaction	−0.06	−0.09, −0.03	0.07	0.04, 0.10	−0.004	−0.007, −0.002	
Autonomy need satisfaction	−0.07	−0.11, −0.03	0.05	0.02, 0.07	−0.003	−0.006, −0.001	
Competence need satisfaction	−0.06	−0.09, −0.02	0.06	0.02, 0.07	−0.003	−0.005, −0.000	
Relatedness need satisfaction	−0.06	−0.10, −0.03	0.05	0.03, 0.08	−0.003	−0.006, −0.001	
Notes.

All path coefficients are statistically significant at the p < .05 level. All coefficients are represented by unstandardized slopes. Our preference was to report standardized coefficients but it is difficult to precisely estimate the confidence interval of standardized indirect effects (Cohen et al., 2003).

Social and physical activity

IGD may displace real-life social and physical activity, and by doing undermine health directly (Sisson et al., 2010) or indirectly by relating to need satisfaction (Mellor et al., 2008; Ng et al., 2012; Wilson et al., 2003). Given this, the mediating effects of psychological need satisfaction might be apparent because need satisfaction correlates with social activity, and these activities provide a better account for the harmful effects of IGD on health than do needs. To test this alternative explanation, we evaluated two additional cross-lagged models that included either social or physical activity (see Fig. 2; Exploratory analyses 5 & 6). Results showed social (β = 0.42, p < .01) and physical (β = .62, p < .01) activity were consistent over time and supported our expectations that physical activity at Time 1 would be associated with later health (β = 0.07, p < .001), although social activity at Time 1 did not relate to health at Time 2 (β = 0.02, p = .21). Contrary to our expectations (Exploratory analyses 7 & 8), those who exhibited more IGD symptoms at Time 1 did not report different levels of social (β = 0.01, p = .68), or physical (β = 0.02, p = .12) activity at Time 2. Thus, we did not find support for indirect effects through these activities. Moreover, links identified between IGD at Time 1 and needs at Time 2, and between needs at Time 1 and health at Time 2 (the two links suggesting the presence of an indirect effect), were significant controlling for activity levels (for the link between IGD and need satisfaction, controlling for physical activity; β =  − 0.06, p < .001, controlling for social activity; β =  − 0.06, p < .001; for the link between need satisfaction and health, controlling for physical activity; β = 0.09, p < .001; controlling for social activity; β = 0.07, p < .001). This constellation of results suggests that the effects of IGD on social or physical activities do not provide a compelling alternative account of why IGD undermines health over time to that offered by need satisfaction.

Discussion

Internet-based games are among the most popular forms of human recreation and empirical research is still needed to understand possible psychopathology related to their use. The present research rigorously investigated the etiology and outcomes of Internet Gaming Disorder and the findings derived from this prospective study inform our understanding of how this phenomenon is linked to dysregulation and health. Guided by an open science approach, results confirmed a number of our preregistered hypotheses concerning dysregulated online play.

In line with predictions we found that the IGD criteria proposed in the DSM-5 (American Psychiatric Association, 2013) were, on an individual and continuous basis, moderately stable over a six month period. Contrary to what we expected, however, none of the participants meeting diagnostic thresholds at the start did so at the end of the study, and only three participants reported more than four IGD criteria at the start and six months later. These findings, that very few, if any, individuals who meet the proposed diagnostic thresholds over time mirror those derived from other large-scale representative studies of problematic gaming research (Festl, Scharkow & Quandt, 2013; Scharkow, Festl & Quandt, 2014). These unexpected results do not support a theoretical framing of Internet Gaming Disorder as a chronic psychiatric condition akin to substance abuse disorder as some have argued (e.g., Hasin et al., 2013; Petry et al., 2014); rather, the constellation of results we uncovered provide evidence that dysregulated gaming is a nuanced phenomenon that requires careful conceptualisation, and one which can be fruitfully studied from a motivational perspective (Deci & Ryan, 2000; Griffiths et al., 2016; Kardefelt-Winther, 2014b). These results may also speak to the nature of the proposed disorder. For example, they mirror some research on problematic gambling, another kind of behavioural dysregulation, which shows such difficulties are more episodic than continuous (Slutske, Jackson & Sher, 2003), though it is unclear whether these IGD episodes are chronic across a span of multiple years, similar to models of addiction where the individual is never truly free of the illness but only experiences intermittent expressions of it (Saitz et al., 2008), possibly as an expression of maladaptive coping (Kardefelt-Winther, 2014a). With this in mind, further research investigating the nature of IGD as chronic or episodic would be useful. Generally, IGD has been measured in terms of its more or less frequent occurrence across a period of 12 months (Pontes et al., 2014), but a lack of stability in clinical thresholds being met across a six-month period suggest that these symptoms would need to be frequently reoccurring over a 12-month period for them to be captured by these longer-term assessments.

Also contrary to our expectations, we did not find that IGD had an observable direct effect on health over time. Although this finding is inconsistent with some results derived from small-scale convenience samples, it is in line with the only other representative longitudinal work which suggests mixed or non-significant lagged effects linking problematic gaming with life satisfaction and perceived success of gamers (Scharkow, Festl & Quandt, 2014). This negative finding is especially noteworthy because it indicates that IGD may not, on its own, be robustly associated with important clinical outcomes. As such, it may be premature to invest in management of IGD using the same kinds of approaches taken in response to substance-based addiction disorders, for example with TMS (Meng et al., 2014; Shen et al., 2016; Terraneo et al., 2016). Further, this pattern of findings suggests that more high-quality evidence regarding clinical and behavioral effects is needed before concluding this is a legitimate candidate for inclusion in future revisions of the DSM-5.

Despite the absence of a direct link with health, additional research findings indicated there is reason for concern when individuals exhibit IGD symptoms. Informed by the human motivation and self-regulation literature on psychological needs (Ryan & Deci, 2000), we predicted and found that those who were not psychologically need satisfied were more likely to evidence symptoms of IGD at a later time. Though the observed relations were not large in magnitude, they suggested that IGD symptoms can emerge from dysregulating environments or dysregulated psychological states brought on by the absence of psychological need satisfaction, in line with other symptoms indicative of psychopathology, such as depression and anxiety (Deci et al., 2001; Talley et al., 2010; Wei et al., 2005), disordered eating (Bartholomew et al., 2011), and borderline personality disorder (Ryan, 2005).

In line with this idea and the research reviewed above, it may also be that endorsing IGD criteria may be characteristic of a broader and more pervasive problem with self-regulation. Indeed, work with individuals who exhibit gambling disorder, the only non-substance related addiction which is recognized in the DSM-5, shows a lifetime prevalence rate of 61% for mood disorders, 75% for alcohol use, and 48% for drug use, rates higher than in the general population and which reflect a history of dysregulation (D. C. Hodgins, Peden & Cassidy, 2005). Given we did not test comorbidity with other clinical disorders, or lifetime prevalence of other clinical disorders in those who exhibit IGD symptoms, future research doing so would greatly enhance our understanding of the nature of the disorder and its treatment.

As expected, we found small yet consistent reciprocal relations existing with needs, such that those who exhibited symptoms of IGD were less likely to be need satisfied later. This observed pattern of joint causality between needs and IGD suggests that the dynamics underlying unhealthy behaviors mirror those observed in other life contexts (Forest et al., 2011). In addition, the confirmation of this hypothesis indicates that dysregulated gaming may be detrimental to experiencing psychological need satisfaction through other avenues and may crowd out more psychologically edifying pursuits (Chen et al., 2015); for example, IGD symptoms may directly interfere with pursuing other meaningful life goals that satisfy needs (Niemiec, Ryan & Deci, 2009). Alternatively or in addition to this, the experience of compulsion and obsession may directly leave individuals feeling that they have less choice, more isolated and lonely, and ineffective (Lalande et al., 2015).

Our data were collected from two time points but with a certain reasonable assumption, we can infer potential mediational process from the cross-lagged model that we tested (Little et al., 2007). Indeed, findings from the prespecified model suggested that need satisfaction mediated the effects of IGD symptoms on health; that is, the results indicated that IGD decreases health through lowering need satisfaction. In other work, such need satisfaction have been shown to relate to the health criteria we have tested in this study; that is, better mental health (Ryan & Deci, 2001), better physical health (Reinboth, Duda & Ntoumanis, 2004; Sheldon, Ryan & Reis, 1996; Thompson & Prottas, 2006), and better psychosocial functioning (H. S. Hodgins, Koestner & Duncan, 1996; Moller & Deci, 2010; Patrick et al., 2007; Weinstein, Hodgins & Ostvik-White, 2011). The current work extends this literature by suggesting that such costs to health are accrued when dysregulating or pathological behaviors undermine need satisfaction. Further, it informs the IGD and the behavioral addiction literatures by highlighting that IGD symptoms lead to lower health partly because they undermine need satisfaction.

Interestingly, in exploratory analyses we found that all three psychological needs served to link IGD to health. Although the three needs are often tested in sum (e.g., La Guardia et al., 2000), additional information can be gained by evaluating their separate impacts on wellness (Sheldon, Ryan & Reis, 1996; Weinstein & Ryan, 2010), and they have been shown to differentially affect psychological outcomes in certain contexts (e.g., Legate et al., 2013). Evidence that each has a direct effect excludes the possibility that, as an example, IGD induces a feeling of loneliness (absence of relatedness need) that is so robust it carries results using the full measure of need satisfaction.

Our final two models assessed whether social and physical activity accounted for the link between IGD symptoms and health, an especially important test given these activities may have been responsible for the indirect effects through psychological needs observed in earlier analyses. The data did not support this conclusion, and instead showed none of the expected relations between IGD symptoms and later costs to social and exercise behaviours. These findings are in line with a mixed literature in this area (Boone et al., 2007), including descriptive research which identifies that 83% of adolescents feel more connected to their friends through their technology use, and suggesting that 68% say they have received social support using technology in tough or challenging times (Lenhart, 2015). It is also consistent with findings that gaming can at times promote physical activity (Lanningham-Foster et al., 2006), and work which fails to identify a consistent relation between gaming and physical activity (Kremer et al., 2014; Mentzoni et al., 2011). In this study, we found that dysregulated gaming did not result in actual social isolation or physical inactivity, and that the subjective experience of psychological need satisfaction continues to link IGD to health even controlling for these activities. Yet it might be that IGD symptoms interfere with other meaningful activities not tested in this research that might undermine need satisfaction and provide a more direct account of why IGD undermines health. For example, as discussed above IGD may interfere with the pursuit of meaningful goals, or with academic or work responsibilities. In future work, researchers may test the possibility that engagement in other daily activities might provide a better account than physical or social activity.

It is important to note that much of the existing literature on IGD has relied on convenience samples of young adult volunteers drawn from online gaming forums (e.g., Pontes et al., 2014). This approach that differs markedly from studying the phenomenon in a representative sample of adults of all ages. Though the former approach presents serious challenges for generalisability and the two populations may be different in terms of how they cope with daily experiences (e.g., Compas et al., 2001; Garnefski et al., 2002), we expect that the mechanisms which we have examined here—namely basic psychological needs to have similar explanatory power in both. This is in no small part because nearly 40 years of SDT research has studied the impact of need satisfaction in diverse populations and across developmental stages in non-digital contexts. Indeed, psychological needs are shown to be comparably important in terms of both self-regulation and coping for children, adolescents, and adults (see review in Ryan & Deci, 2017).

Limitations

The present research has a number of limitations which future research can address to improve our empirical understanding of self-regulation and IGD. First, the study presented relied on self-reports provided at two time points. The use of respondents for explanatory and criterion data collected misses the opportunity for considering converging and diverging information from the broader social context (Campbell & Fiske, 1959; Cronbach & Meehl, 1955), and the inferences we draw in these structural models rests on an assumption of equilibrium. Given the reliance on a brief self-report measure, we may not have fully examined the construct of psychological distress given the clinical framework conceptualizes this as “clinically significant impairment or distress.” In future research both aspects of this criterion—that is, significant impairment and distress—must be evaluated carefully and separately. For example, it might be that an individual with IGD does not experience distress, but his or her game playing disrupts academic or professional performance in a meaningful way. Previous research has shown that such questions may vary in suitability as self-reports—that is in some cases but not all individuals are able to make accurate self-assessments (Clancy & Gove, 1974; Murphy & Schachar, 2000; Weiss et al., 1998). Future studies could therefore collect reports of health and behaviour from friends, family, or caregivers as well as direct behavioural observations of gaming patterns using gaming logs and collected data at a greater number of time intervals so that an equilibrium assumption can be relaxed in modeling to infer mediation processes.

Second, the present data cannot speak to the motivation dynamics of the specific games participants played (e.g., Przybylski, Rigby & Ryan, 2010). Given that the current data suggest motivational factors such the absence of basic need satisfaction is part of problematic play, future studies of Internet Gaming Disorder should examine motivational factors outside of and within games. Third, the study focused on a sample of American adults. Particular concerns have been expressed about the potential effects of Internet gaming on cohorts we did not study, including young people and those living in Eastern societies (American Psychiatric Association, 2013). Our sample of individuals from the general population may not have been sensitive to the dynamics underlying extreme and rare cases of problematic gaming, which may show a different pattern of play (for example, these players may exhibit a more continuous nature to their disordered play). Future studies could examine the health and dysregulated play correlates of IGD in samples of younger participants such as school aged children and in other cultural milieus such asEastern societies where the social nature of online games differ. Fourth, the study design involved a six-month long latency between measures, which allowed for testing lasting effects but may not have been particularly sensitive to immediate causal effects on or by IGD. This is especially problematic if the nature of the disorder involves short-term increases and declines in symptoms over a period of days. Future studies testing the model across varied latencies will be important, for example using daily diary methodology that is more sensitive to experiences as they occur (Wheeler & Reis, 1991).

In addition, despite recruiting a large sample, the number of participants endorsing multiple IGD criteria was quite low. This low prevalence limits the complexity of models that can be applied to the data at this scale, and is a generally unacknowledged problem in the field (for an exception see: Festl, Scharkow & Quandt, 2013). Given that resource intensive large-scale data are needed to accurately study the phenomenon, we strongly suggest that researchers pool their efforts and openly share their data, code, and materials. This will limit duplicate and divergent efforts arising from less systematic approaches. Finally, the study only examined links between IGD and motivational and health outcomes. If indeed IGD is a pervasive or special case of behavioural dysregulation it should have a material and pervasive influence on self-control in other domains (Chen et al., 2015; Deci & Ryan, 2000; Steel & Blaszczynski, 1998). Therefore, future studies could also examine the relations between IGD and psychosocial functioning in school, work, and other recreational contexts.

Conclusions

The possibility that future revisions of the Diagnostic and Statistical Manual of Mental Disorders may codify Internet Gaming Disorder as a psychiatric condition meriting clinical attention and resources is controversial (Aarseth et al., 2016). Given the high professional and reputational stakes, studies purporting to investigate the phenomenon must complement robust theory with transparent scientific practices. The present work represents our concerted effort to this end. We used self-determination theory as a lens to study the need and health consequences of IGD and utilized a robust open-science empirical approach. Our use of SDT, open materials, open data, and clearly distinguished confirmatory hypotheses (i.e., pre-registered) and exploratory analyses informs the existing literature. By building on this foundation, our findings speak meaningfully to the etiology and health impacts of IGD. They suggest that IGD has dynamics of dysregulations brought on by the absence of psychological need satisfaction in one’s environment, and that it has implications for health, but only by undermining need satisfaction further.

This work also provides some data to guide future work in this area. For example, we found evidence that individual IGD symptoms may remain stable over a brief period of time, yet individual diagnoses appear not to be. Support for the basic psychological needs for competence, autonomy, and relatedness lead to declines in IGD over time, though we did not observe direct effects of IGD on health over time. Tentative evidence consistent with motivational theory suggested IGD may have an indirect effect on health insofar as it undermines need support, yet further research is needed to determine the conditions underwhich such links endure. Only when such steps are taken will we know if the attention that researchers and clinicians give to the potential darker aspects of this immensely popular Internet-based activity is fully justified.

Additional Information and Declarations

Competing Interests

Author Contributions

Human Ethics

Data Availability

The authors declare there are no competing interests.

Netta Weinstein, Andrew K. Przybylski and Kou Murayama conceived and designed the experiments, performed the experiments, analyzed the data, contributed reagents/materials/analysis tools, wrote the paper, prepared figures and/or tables, and reviewed all drafts of the paper.

The following information was supplied relating to ethical approvals (i.e., approving body and any reference numbers):

Ethical review for data collection and analysis was conducted by the Research Ethics Committee at the University of Oxford (C1A15006).

The following information was supplied regarding data availability:

Przybylski AK, Murayama K, Weinstein N. (2017) Prospective Study of Internet Gaming Disorder in a U.S. Cohort. http://dx.doi.org/10.17605/OSF.IO/GE9TP.

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
