# Peer review of "A prospective study of the motivational and health dynamics of Internet Gaming Disorder"

_PeerJ, doi:10.7717/peerj.3838_

## Round 0.1 · original submission · Minor Revisions

You have received 4 thoughtful, positive reviews. Please revise your manuscript accordingly.

Reviewer 1 ·

Basic reporting

The present study recruited 5,777 American adults and applied self-determination theory to examine how motivational factors influence, and are influenced by, IGD and health across a six month period. The sample and the longitudinal design is very impressive. Overall, it is a very important article for this filed.

Experimental design

Research question well defined, relevant & meaningful. The authors should take "internet addiction test" into their consideration for the examination for the participants if they collected this score.

Validity of the findings

Data is robust, statistically sound, & controlled.

Additional comments

The present study recruited 5,777 American adults and applied self-determination theory to examine how motivational factors influence, and are influenced by, IGD and health across a six month period. The sample and the longitudinal design is very impressive. Methods described with sufficient detail & information. The results were robust and interesting. Overall, it is a very important article for this filed.

Reviewer 2 ·

Basic reporting

The current study offers a substantial advance in our knowledge and understanding of a complex and controversial disorder. Furthermore, the authors should be commended for taking an open science approach to the study, and pre-registering their analyses. In sum, I believe that this paper has a vitally important contribution to make to the IGD literature – not just in terms of the findings, but also as a model example of the way psychological experiments should be conducted.

Experimental design

The authors present here a large-scale study looking at the presence of Internet Gaming Disorder (IGD) symptoms in a substantial sample of American participants, and apply self-determination theory to assess if and how presence of IGD affects health and behavioural outcomes over a six month period. The research is robust, the research questions are well-defined, and fill an important gap in the research literature.

Validity of the findings

The study presents some fascinating and novel findings. IGD appears to be disruptive of some health factors in terms of psychological need supports, but the authors present evidence of no direct effect of dysregulated gaming on health over time. While IGD symptoms appear to be moderately stable over the six month period, the authors in fact show evidence of a drop in problematic symptoms over that time – no one who met a diagnostic threshold at the start did so at the end. This finding has important consequences for the theoretical framework within which we consider IGD – currently, there is some debate as to whether it should be framed as either a substance abuse disorder, or as more of an impulse control disorder. Before we can make any headway in developing effective ways of assessing and treating IGD, this debate first needs to be settled.

·

Basic reporting

The most important concern is the clarity of presentation of the many analyses. There are only two tables and two figures for the many analyses presented, and statistical estimates are presented in only one of these. Path analyses and SEM models are often reported with statistical estimates included. If possible, reporting the various path analyses in clearly-labeled and described figures would be useful. The reporting of fit statistics would be useful as well. Although the main model is described as a fully saturated model, the wording of that sentence (335) may make it appear as if this is a result the fit of the model rather than a property of saturated models by definition. This line would benefit from being reworded. Further discussion of modeling is covered in the Validity section.

Second, there are some instances of terminology that were unfamiliar or required some thought/research to understand. While psychological need satisfaction is well-known. I was unable to find many references to "need supports". For example, it appears in work by Deci and Ryan, but not in the works cited in the paper (around line 91). In addition, use of the term "need supports" in a sentence is a little confusing because the word "supports" is more often used as a verb. For these reasons, I would suggest the authors consider using the term “need satisfaction” or even reword sentences containing “need supports” to improve clarity. Likewise, “motivational dynamics” is another term that may not be easily understood outside this field. While these terms are certainly relevant and accepted within certain areas of study, the paper would be better understood by those outside of these areas if other language is used.

Experimental design

This paper uses a design common to survey studies. Any concerns are more related to reporting and validity and are covered in those sections.

Validity of the findings

One of the major strengths of this paper is the potential representativeness of this sample. However, external validity is a concern. Some properties of the sample make it seem less representative than expected. The mean age is 46, which seems high for an Internet gaming sample. Without performing calculations, it is difficult to tell the racial makeup of the sample, which is another important factor in determining representativeness. There is no information on socioeconomic status. The sample has a higher proportion of females than males; is that in line with expectations for a sample of Internet gamers? In understanding how the important results of this study may relate to other populations of Internet gamers (e.g., younger, mostly male), it is important to have a little more explanation about the intent, recruitment and data collection in the original study and a critique of how the study sample may differ. A final note about the study procedure: the method of informed consent should be specified. What does it mean that there was a double opt-in?

The time period of the analysis is also a consideration, especially as it relates to the sample’s age. Would a 6-month period be adequate to capture causal effects, especially if the disorder itself is short-term in nature? Some mention of this potential problem would help contextualize the findings.

The reporting of analyses also brings up questions of validity. As stated above, the saturated model is described as having perfect fit, but this tells us nothing about the meaning or potential invalidity of the estimates as it is a property of a saturated model. Several methods can be used to evaluate the appropriateness of such models (see a standard text on structural equation modeling such as (Kline, 2010). If I understand correctly from the code, a multigroup analysis by sex was conducted. This would be an example of an alternative model that could be tested and reported. Another concern is how the use of FIML was conducted. More explanation of the steps involved in lavaan would be useful for those who are less familiar with this package. For example, the authors could specify, “The full information maximum likelihood estimator was used in the lavaan package in R by …”

A final concern major consideration is the wording of one of the IGD indicators. Starting on line 255, the paper discusses the “distress” criterion of IGD. In fact, the wording of the IGD criterion is “clinically significant impairment or distress”. The self-report of distress may differ from the self-report of impairment, and the importance of this clinical judgment is, of course, a debate in the field. I would be especially interested to know if self-report of distress differed by sex or age group, for example.

Additional comments

This paper reports a very thorough analysis of representative longitudinal data on psychological needs, Internet gaming disorder and health using a well-researched and thoroughly developed rationale. The strengths of the study are its clearly-defined hypotheses, its transparency, and the connection of hypotheses and results. As the Introduction states the intent to bridge the gap in the literature between self-regulation and addiction, this reviewer reports from the perspective of a researcher trained in public health and addictions with suggestions for reporting that will make the strengths of the article more accessible to those in the biomedical sciences.

Overall, the authors have conducted a well-thought out study that would benefit from some improvements in reporting to clarify limitations related to sample and to the complex analyses. I have outlined other concerns or provided more information about the items summarized above in a line-by-line analysis below. It is certainly not necessary to respond to each line, however.

I welcome the chance to be involved in reviewing a revised version of this manuscript.

Line by line review
Line 64: Individual indicators in line with APA guidance- The word indicators has a specific statistical meaning. It would be better to use criteria (as they are referred to in the DSM) or symptoms. The word appears again in line 245 and other places.

Line 76: The reference provided refers to a very specific population, and thus may be overstated. Since there is also good support for the opposite (i.e., detrimental effects on populations) ,it would be better to find additional support for that conclusion (e.g.,(Hofferth & Moon, 2012)) and to critique studies that find that time gaming is associated with harm at the population level (e.g., (Faulkner, Irving, Adlaf, & Turner, 2014; Festl, Scharkow, & Quandt, 2013). If the concern is a lack of longitudinal studies that provide additional support for causal determination, that also has conflicting evidence in the literature (e.g., (Gentile et al., 2011). With regard to the Scharkow et al. study in 2014, it’s also noteworthy that no longitudinal associations between PG and well being were found for adolescents.

Lines 78-82: These lines seem to be making the case that PG is caused by dysregulation, but the statement on line 82 is that it is “dysregulating”, implying that it causes dysregulation.

Line 91: I was initially confused by the term psychological need supports, as I have never seen this term in the biomedical literature. I did make a point of reviewing some of the cited articles, and I see that they use other terminology such as need satisfaction. I would recommend that if this article is meant to bridge the literature, terminology that is more straightforward would be easier to understand. Line 101 is another example of where the terminology makes the sentence a little confusing. Some alternatives might be “need satisfaction”, which seems more common in the SDT literature, or even rewording to remove jargon. For example, line 99-100 could be worded as “Having psychological needs satisfied is associated with better mental, physical and psychosocial health.”

114: A similar problem is reflected here, where the claim that Psychological Needs Reduce Dysregulation is confusing. I think it is having psychological needs met that reduces dysregulation.

144-147: In discussing the displacement of activities, this statement seems out of place.

155: Obsessive passion for video games? Internet games? Activities in general?

157: Gaming and Everyday Activities. While the previous section uses an SDT framework to discuss the potential mechanisms of effects of engagement on psychological needs, the section wants a framework. The authors might consider mentioning the displacement hypothesis (Valkenburg & Peter, 2007). Another concern with this section is that it discusses the importance of social activities without providing a reason that reduced social activities might be associated with health. Social participation is a basic function of everyday life, and theories such as the stress buffering hypothesis (Cohen & Hoberman, 1983) might help support the importance of social activities for health.

175: Motivational dynamics is another term that is somewhat unclear at first.

220: From a public health perspective, it would be important to have more information about the sample and study. For instance, how was the survey conducted? Was it conducted online, by telephone or in person? How did the sample of Internet gamers compare to the overall sample? I notice that the average age is 46.2; how does this compare with studies using other population samples in this topic area? The reporting of race is concerning, as it is difficult to tell the representativeness without performing the necessary calculations to determine what percentage of the sample is White (2234/3146). Percentages should be included here.

235-243: Was informed consent provided, or was the double opt-in considered informed consent in some way?

255: The wording of the proposed IGD definition is “ impairment or distress”. Obviously the impairment piece is another point of contention in the IGD debate, especially as the reliability of respondents to truthfully report impairment is called into question for any mental disorder. The paper should address the discrepancy between the wording for this survey item and the requirement for “clinically significant impairment or distress” in the proposed IGD diagnosis.

279: Please include a sample question for those who are new to these scales.

328: I’m confused as to how the inclusion of T1 IGD scores contributes to FIML estimation when these variables are a part of the specified model. Perhaps the sentence needs to be reworded to describe which other variables, if any, were also included that were associated with attrition.

335: Reporting of model fit. The results section describe the model as saturated and producing perfect fit with d.f.=0. Since saturated models by definition produce perfect fit, it is necessary to use other methods to assess whether the fit is in line with expectations. The authors might consider conducting hypothesis testing comparing their chosen model with other potential models, for example.

353: Small significant betas may result from the large sample size. It is good that they were consistent with the author’s predictions in terms of being in the appropriate direction, but what is the theoretical implication of a beta of magnitude 0.06? Expressing standardized betas in the form of change in expected outcome for standard deviation of the chosen indicator would make this more clear. (e.g., for every standard deviation change in X, Y was expected to increase by ___) It also seems that at times that unstandardized betas are reported (e.g., Table 2). This would be good to clarify with each set of models discussed, or report one or the other consistently. When development of a construct is the question, unstandardized estimates, including residual covariance, allow for a better understanding of heterogeneity in change and the ability of the model to explain relationships. This is another potential consideration.

390: I’m not sure, but it seems that “need supports correlates” might be a typo.

409: It doesn’t seem to me that 10+ years of the study of problematic online gaming (e.g (Kraut & Seay, 2007)), would qualify for being in the early stages. That’s not to say that the current state of the evidence is enough to correctly conceptualize a formal disorder. Nonetheless, I think it’s an overstatement to say that work is in an early stage.

Beginning on line 415: This paragraph has very important implications for IGD and similar diagnoses.
The finding that a diagnosis of IGD based on satisfying a number of criteria does not hold at 6 months implies that the 12-month time criterion may lead to brief periods of “disorder” being missed. Have there been other suggestions as to a relevant time period for IGD? This would be important to know and discuss. Also, the paper describes options for time periods of addictive disorders are chronic or episodic. From a more medical perspective, addictions are often seen as an “absorbent state” (as are other mental disorders) where individuals never truly recover, but suffer from occasional episodes. Problems related to gaming might also represent a phase, however, which is more in line with the maladaptive coping hypothesis (Kardefelt Winther, 2014). In this case, rather than chronic or episodic, quickly resolving might be another possible way to describe problems. The potentially brief nature of disordered gaming also has implications for the 6-month lag in the current study. The authors are encouraged to discuss the implications for their finding in terms of the potential ephemeral nature of online gaming-related problems, how their sample may have influenced this finding in their analysis, and also how these may differ by populations.

470: I was not familiar with the word choiceful and had to look it up. I see it is British English, and while this is of course perfectly acceptable, I am still in favor of rewording to make it even more clear.

483: The paper states that IGD symptoms are linked to health “partly because they undermine need supports”, yet previous it seems that IGD symptoms were only indirectly linked to health. Some way to organize results more clearly (additional tables with clear description of models tested, e.g.) would be useful.

498-501: The literature cited here refers to samples of adolescents, yet the sample in the study is of middle-aged adults. The authors should consider how mechanisms may differ in their sample.

555-556: “absence of supportive environments” has a much different connotation in the public health literature, speaking mainly to child development and a life course approach. My understanding is that motivational dynamics as studied here may the result of environments experienced during upbringing, but are not necessarily limited to this. That phrase may benefit from being reworded.


Figures: The results of the hypothesized model (the betas) would be much easier to understand if they were indicated in the path model rather than solely in the text.

Figure 1 note: It is unclear which nested models were tested and how covariances were evaluated.

Reviewer 4 ·

Basic reporting

This manuscript describes a study of the motivational and health dynamics of Internet Gaming Disorder. The task was well controlled. The study applied self-determination theory to examine the relationship of IGD, motivational factors and health across a six month period. The results confirmed that IGD indicators are stable and that they and basic psychological need supports have a reciprocal relationship over time. The need supports promoted health and served as a protective factor against IGD. This is an important topic and the question is of wide interest for researchers working in IGD.
In discussion relate to Internet addiction management, I will recommend to include 10 Hz rTMS can be used to treat addiction. Reference (1): Shen et al. 10-Hz Repetitive Transcranial Magnetic Stimulation of the Left Dorsolateral Prefrontal Cortex Reduces Heroin Cue Craving in Long-Term Addicts. Biol Psychiatry. 2016 Aug 1;80(3):e13-4.
(2) Transcranial magnetic stimulation of dorsolateral prefrontal cortex reduces cocaine use: A pilot study. Eur Neuropsychopharmacol. 2016 Jan;26(1):37-44.
In addiction, tDCS helps to treat addiction.
Reference: (1) Meng et al. Transcranial direct current stimulation of the frontal-parietal-temporal area attenuates smoking behavior. J Psychiatr Res. 2014 Jul;54:19-25
(2) Wang et al. Transcranial direct current stimulation of the frontal-parietal-temporal area attenuates cue-induced craving for heroin. J Psychiatr Res. 2016 Aug;79:1-3.

Experimental design

This study has clear description in the material and methods. The work conduct in conformity with the ethical standards of the field.

Validity of the findings

The statistical methods are used valid in this study.

---

## Round 0.2 · accepted · Accept

Thanks for your efforts on revising the paper.